# Delivering maternal and childcare at primary healthcare level: The role of PMAQ as a pay for performance strategy in Brazil

Olívia Lucena de Medeiros[1,2], Jorge Otávio Maia Barreto[1,3], Matthew Harris[4], Letícia Xander Russo[5], Everton Nunes da Silva[1]*

1 Graduate Program in Public Health, University of Brasilia, Brasília, Brazil, 2 Ministry of Health, Brasília, Brazil, 3 Oswaldo Cruz Foundation, Rio de Janeiro, Brazil, 4 School of Public Health, Faculty of Medicine, Imperial College, London, United Kingdom, 5 Department of Economics, State University of Maringá, Maringá, Brazil

* evertonsilva@unb.br

**Data Availability Statement:** All relevant data are within the manuscript and its Supporting Information files.

## Abstract

### Background

Improving access and quality in health care is a pressing issue worldwide and pay for performance (P4P) strategies have emerged as an alternative to enhance structure, process and outcomes in health. In 2011, Brazil adopted its first P4P scheme at national level, the National Programme for Improving Primary Care Access and Quality (PMAQ). The contribution of PMAQ in achieving the Sustainable Development Goals related to maternal and childcare remains under investigated in Brazil.

### Objective

To estimate the association of PMAQ with the provision of maternal and childcare in Brazil, controlling for socioeconomic, geographic and family health team characteristics.

### Method

We used cross-sectional quantile regression (QR) models for two periods, corresponding to 33,368 Family Health Teams (FHTs) in the first cycle and 39,211 FHTs in the second cycle of PMAQ. FHTs were analysed using data from the Brazilian Ministry of Health (SIAB and CNES) and the Brazilian Institute for Geography and Statistics (IBGE).

### Results

The average number of antenatal consultations per month were positively associated with PMAQ participating teams, with larger effect in the lower tail (10th and 25th quantiles) of the conditional distribution of the response variable. There was a positive association between PMAQ and the average number of consultations under 2 years old per month in the 10th and 25th quantiles, but a negative association in the upper tail (75th and 90th quantiles). For the average number of physician consultations for children under 1 year old per month, PMAQ participating teams were positively associated with the response variable in the lower tail,

**Funding:** MH is supported in part by the NW London NIHR Applied Research Collaboration. Imperial College London is grateful for support from the NW London NIHR Applied Research Collaboration and the Imperial NIHR Biomedical Research Centre. The views expressed in this publication are those of the authors and not necessarily those of the NIHR or the Department of Health and Social Care. No additional external funding received for this study.

**Competing interests:** The authors have declared that no competing interests exist.

but different from the previous models, there is no clear evidence that the second cycle gives larger coefficients compared with first cycle.

## Conclusion

PMAQ has contributed to increase the provision of care to pregnant women and children under 2 years at primary healthcare level. Teams with lower average number of antenatal or child consultations benefited the most by participating in PMAQ, which suggests that PMAQ might motivate worse performing health providers to catch up.

## Introduction

Improving access and quality in health care is a pressing issue worldwide, particularly in low- and middle-countries [1]. Pay for performance (P4P) strategies have emerged as a policy option to enhance the structure, process and outcomes of healthcare [2]. P4P strategies are often adapted to specific country needs [3] and can include several targets and levels of care. Although the implementation of P4P has increasingly been implemented since the 1990s [4, 5], there are few rigorous P4P evaluations, and overall evidence of its effects is weak or mixed [6, 7]. There are also ethical issues and concerns regarding its impact on deprived population groups leading to worsening health inequalities. P4P programmes oftentimes focus only on process and structural indicators without measuring the effect on outcomes, such as mortality [8].

In 2011, Brazil launched the National Programme for Improving Primary Care Access and Quality (PMAQ, acronym in Portuguese), which is a P4P programme with the aim of increasing access and quality of care at the primary care level across Family Health Teams (FHTs). FHTs provide coordinated, comprehensive, and continuous care through multidisciplinary teams composed of physicians, nurses, nurse assistants and community health agents. FHTs include teams from the Family Health Strategy (FHS) and Primary Health Teams (PHT). The difference between FHS and PHT refers to team composition (hiring health community agents and nurse assistants are not mandatory for PHT) and nurses' and physicians' workload (higher for PHT). PHT represents less than 1% of the total FHT in Brazil. The PMAQ involved bonus payments provided from the federal government and are based on performance against agreed health indicators. These are verified by means of administrative records, self-assessment of the participating teams, and the results of an external assessment carried out by universities [9]. The PMAQ bonus ranges from R$ 1,700.00 (US$ 434.55) to R$ 11,000.00 (US$ 2,811.79) per team/month based on their performance. The participation in PMAQ is voluntary for municipalities, however it has expanded rapidly, reaching 51.4% (first cycle, in 2011/2012), 77.6% (second cycle, in 2013/2015) of the total number of PHTs in Brazil [10]. The third cycle started in 2016, reaching 93.9% of the teams, and due to delays in conducting the external evaluation the final performance score was launched at the end of 2018 [11, 12]. PMAQ is funded by the Ministry of Health (MoH), which has invested R$ 9.85 billion (US$ 2.52 billion) from 2011 to August 2018 [13, 14].

Although PMAQ has a vast set of agreed health indicators in terms of access and quality, in this study we focus on the indicators that relate to the delivery of maternal and childcare services. Provision of maternal and childcare is an international concern and a focus of the United Nations Sustainable Development Goals [15]. Brazil has progressed in this field, particularly due to higher coverage of primary healthcare facilities and improved socioeconomic

development in the last decades [16]. However, some challenges remain. A nationwide hospital-based study carried out with 23,894 pregnant women in 2011 and 2012 showed that only 21.6% had overall adequate antenatal care. This was defined as starting no later than the 12th gestational week, with at least six consultations, a record in the antenatal card of at least one result for each of the recommended routine antenatal tests, and guidance regarding the maternity hospital for delivery [17]. Results from the Birth in Brazil Survey showed that inadequate antenatal care was strongly associated with neonatal death [18]. Moreover, Brazil has recently adopted fiscal austerity policies associated with an increase in child morbimortality [19].

The published literature on PMAQ has increased during the last years, but studies largely rely on descriptive analyses [20]. On this basis, our objective is to estimate the impact of the PMAQ on the provision of maternal and childcare in Brazil, controlling for socioeconomic, geographic and team characteristics. A quantile regression approach was used because it enables a more detailed mapping of the association of the exposure variable (PMAQ participation) and covariates (socioeconomic, geographic and team characteristics) on the conditional distribution of response variables, which in this study were the average number of antenatal consultations per month, the average number of consultations for childcare under 2 years old per month and the average number of physician consultations for children under 1 year old per month. These outcomes were chosen for clinical reasons, as these consultations are essential procedures for good practice in maternal and childcare, if delivered in a timely and high-quality manner [21–23]. For example, in sub-Saharan African countries, there is a 39% lower risk of neonatal mortality among women who attended at least one antenatal care visit during pregnancy [24]. In Brazil, 40% of the avoidable neonatal deaths can be attributed to poor antenatal care [25]. Additionally, three consultations in the first year of life reduces the risk of neonatal mortality by 20.5% in Brazil [26].

## Method

### Study setting

Brazil has established the Unified Health System (SUS, acronym in Portuguese), in 1988, providing comprehensive, universal preventive and curative care through decentralised management and provision of health services, and free of charge at the point of service [27]. Primary healthcare was nationally implemented in 1994, and since then the FHS has reached 64.5% of the Brazilian population [28]. The FHS deploys interdisciplinary healthcare teams, which are composed of at least a physician, a nurse, a nurse assistant and four to six community health agents [29]. The FHS can also delivery other services such as: i) dental care, including dentist and dental technicians; and ii) Family Health Support Unit (NASF, acronym in Portuguese), that includes community mental health professionals (psychologists, occupational therapists and psychiatrists), rehabilitation professionals (physiotherapists, speech therapists, nutritionists), as well as paediatricians, gynaecologists and social workers [30]. When all these teams work together in a multidisciplinary way, a more comprehensive care can be delivered to pregnant women and infants. The prevalence of postpartum depression among healthy mothers without a prior history of depression is 17% [31]. NASF can also hire nutritionists, who can integrate diet and nutrition into paediatric primary care, preventing weight gain and obesity [32]. Oral health is also part of overall health. Pregnant women are more likely to have oral problems due to various hormonal changes and fluctuations in intraoral flora; around 40% of pregnant women suffer from some form of periodontitis and up to 10% may develop pregnancy oral tumours [33].

PMAQ underwent several changes over the three cycles, particularly in terms of maternal and childcare. The outcomes used in our model (consultations for antenatal, children under 1

and children under 2) were removed from the third cycle. They are no more incentivized. There is only one indicator related to maternal and childcare in the third cycle: "percentage of new-borns that received care in the first week of life". Moreover, the information system used by the MoH to verify the production of each team also changed (SIAB in the first two cycles and e-SUS in the third one). There is no comparable data between first/second cycles and third one related to maternal and childcare. Based on that, we decided to not include the third cycle in our study.

## Study design

Cross sectional quantile regression (QR) models were employed to investigate the association of PMAQ with the provision of maternal and childcare at primary care level in Brazil. We chose the FHT as the unit of analysis because it is the level where performance is measured by the MoH to pay the financial incentive (bonus payment) for municipalities where the participating PMAQ teams are localised.

We ran the regression models for two periods representing the first and second cycles of PMAQ. The first cycle lasted 18 months, starting on November 2011 (first bonus payment) and ending on April 2013 (last month payment). We opted to use the same length for the second cycle, which covers from May 2013 to October 2014.

## Data source and variables

We have used three sources of information to conduct our study. The first was the Information System for Primary Healthcare (SIAB, acronym in Portuguese), by which we collected data on process indicators related to maternal and childcare, namely the average number of antenatal consultations per month, average number of consultations for childcare under 2 years old per month and average number of physician consultations for children under 1 year old per month. These process indicators were the outcomes of the regression models. It is worth noting that childcare includes consultations for routine growth and development monitoring. Acute care and immunization were not included in our model because they were not available at FHT level. Additionally, the outcomes of childcare visits for children under 1 year and under 2 may reflect overlap. Based on that, we used them in separate models to avoid collinearity. We decided to include both variables because consultation under 1 year are performed exclusively by physicians and consultations under 2 year are performed by physicians and nurses.

We also used an index calculated by the MoH, by which teams are grouped into several strata of similar socioeconomic and demographic status. Regarding socioeconomic indicators, the MoH took into consideration GDP per capita, percentage of population using private health insurance, percentage of population receiving conditional cash transfers and percentage of population living at extreme poverty; and the demographic indicator was population density [34]. The SIAB database was obtained from the MoH. For a full description of the variables, see S1 Table.

The second source of information was the National Register of Health Establishments from DATASUS (CNES, acronym in Portuguese) [35], by which we collected data from individual characteristics related to the FHTs in terms of type of team (PMAQ, dental care and NASF), area where the team is localised (rural or urban) and the number of working-hours available by each health professional of the team (physician, nurse, nurse assistant, community health agent, dentist and dental technician). We included working-hours in the model because it is possible that teams could enhance better results by hiring more health professionals, keeping

work productivity at a constant. Better results would not reflect better effort or commitment with targets.

The third source of information was the Brazilian Institute of Geography and Statistics (IBGE, acronym in Portuguese) [36], by which we collected data on geographic population size and region of the municipality where the team is localised.

## Participants

SIAB and CNES provide team-level data, which were linked based on three key-identifiers: IBGE code of the municipality; health unit code; code of team area. We also used the national team identifier (INE, acronym in Portuguese) to increase the precision of the linkage. We considered all FHTs across Brazil. The data obtained from the MoH correspond to 33,368 FHTs in the first cycle and 39,211 FHTs in the second cycle. We identified incomplete reporting of some teams. Missing data were found in the outcome variables and characteristics of the teams. As these variables are relevant to the analysis, we chose to exclude all observations with missing values. We recorded a loss ranging from 18% to 20% of the observations in the cycle 1 and a loss ranging from 15 to 17% of the observations in the cycle 2, depending on the outcome variable. Missing data is a concern especially for inflexible multivariable methods [37]. In addition to the main analysis, we employed the method for multiple imputation and re-estimated the models as a form of sensitivity analysis. This is a well-known approach for dealing with the loss of observations, with the missing values imputed based on an iterative method.

## Statistical analysis

Descriptive analyses were provided based on mean and standard deviation (SD) of variables used in the regression models for each PMAQ cycle investigated (1st and 2nd). Cumulative distribution, which shows the distribution of the data that has values less than or equal to the quantile values (10th, 25th, 50th, 75th and 90th), was performed by PMAQ and non-PMAQ teams in both cycles. We used the box plot to display the distribution of data, since it provides a useful way to visualise the lower adjacent value (minimum score), lower quartile, median, upper quartile and upper adjacent value (maximum score).

Quantile regression was applied to estimate the association between the provision of maternal and childcare and PMAQ. We compared changes in the number of maternal and child consultations in teams where PMAQ was implemented relative to teams where PMAQ was not implemented, controlling for other covariates. Our empirical specification of the model is described as follow:

$$Q_y(\tau|X) = \beta_{0(\tau)}cons + \beta_{1(\tau)}PMAQ + \beta_{2(\tau)}FHT + \beta_{3(\tau)}SC + \beta_{4(\tau)}GC$$

Where $\tau$ indicates that the parameters are for a specific $\tau$ quantile. $Q_y$ were the outcomes, which represent three different measures of maternal and childcare, namely the average number of antenatal consultations per month, average number of consultations for childcare under 2 years old per month and average number of physician consultations for children under 1 year old per month. We run separate models (regressions) for each outcome. *PMAQ* was our exposure variable, representing teams that were enrolled in PMAQ. *FHT* is a vector of variables representing FHT characteristics. Teams can delivery other primary care services such as dental care and NASF. We also controlled for the working-hours offered by the teams, stratified by health professionals such as physicians, nurses, nurse assistants, community health agents, dentists and dentist assistants.

Socioeconomic status (*SC*) was included into the model by considering six strata calculated by the MoH by which municipalities were grouped based on similar economic environment,

where SC1 represents the poorest municipalities and the SC6 represents the richest municipalities (chosen to be the reference case). As there are several inequalities among the Brazilian regions, we also control for a range of geographic characteristics (*GC*). We included a dummy for the regions North, Northeast (chosen to be the reference case), Midwest, South and Southeast. Rural represents a dummy for teams that delivery primary healthcare services at rural areas. We also controlled for the population size of municipality, which we grouped into three dummies as small (under 10,000 inhabitants), medium (between 10,001 and 100,000 inhabitants) and large (above 100,000 inhabitants, which was chosen to be the reference case).

Quantile regression (QR) is defined by minimising the weighted sum of the absolute values of the residuals [38] and has a number of advantages over ordinary least squares (OLS): it is more robust to outliers; it uses all the data to estimate the quantiles; it is well known that when asymmetries and heavy tails exist, the sample median (the 50th percentile) provides a better summary of centrality than the mean; it enables more detailed mapping of the effect of the covariates on the conditional distribution of a response variable [39–41]. For comparison, we also estimated the coefficients based on an OLS approach.

The statistical analysis was carried out using the software STATA/SE version 14. Data from CNES and IBGE came from publicly available sources. SIAB data were requested to the MoH, and individuals were not identified. According to the Resolution number 510/2016, from the National Commission for Research Ethics (CONEP) in Brazil, research based on database, whose information is aggregate and with no possibility to identify any individual, is not necessary to submit to a research ethics committee.

## Results

PMAQ participating teams presented higher average number of consultations per month related to maternal and childcare in both cycles (1st and 2nd), except for the average number of consultations for childcare under 2 years old where non-participating teams had higher number of consultations in the first cycle (Table 1). The supply of other services at primary health care (dental care and NASF) were more frequent in PMAQ participating teams, and this difference increased in the second cycle. In terms of workforce, only community health agents were more available in non-PMAQ participating teams in terms of working-hour per week. There is no clear pattern for socioeconomic status of the municipality where the teams are located. In the first cycle, 58.9% and 53.8% of the PMAQ and non-PMAQ participating teams were located in the richest municipalities (groups 4, 5 and 6), respectively; these percentages changed in the second cycle to 57.6% and 62.9%, respectively. In relative terms, there were more non-PMAQ participating teams in rural areas (1st and 2nd cycles), North (1st and 2nd cycles), Northeast (1st and 2nd cycles) and medium (1st cycle) and large (2nd cycle) municipalities (Table 1).

We provided cumulative distribution of the average number of consultations per month performed by PMAQ and non-PMAQ participating teams during cycles 1 and 2 (Fig 1). We also compared these values with the MoH recommendations (red line). The upper adjacent line represents the quantile value. Our findings showed that PMAQ participating teams had better performance in maternal and childcare indicators related to non-PMAQ participating teams in both cycles. The differences between the teams have increased substantially for all indicators in the second cycle. Considering antenatal consultations, around 25% of both teams did not achieve the MoH recommendation of 0.77 consultations per month in cycle 1, representing 7 antenatal consultations per pregnancy [21]. In the second cycle, only PMAQ teams performed better, 19.4% of PMAQ teams and 33.4% of non-PMAQ teams had not achieved the recommended number. In terms of consultations for childcare under 2 years, around 57%

**Table 1. Descriptive statistics of the variables used in the model, stratified by participation in PMAQ and cycles, Brazil, 2011–2014.**

| Variable | Cycle 1 | | Cycle 2 | |
|---|---|---|---|---|
| | PMAQ participating (13,971) | Non-PMAQ participating (15,017) | PMAQ participating (28,247) | Non-PMAQ participating (8,142) |
| **Outcome (per month)** | | | | |
| Average number of antenatal consultations | .984 (.385) | .973 (.393) | 1.061 (.394) | .907 (.459) |
| Average number of consultations for childcare under 2 years old | .391 (.297) | .409 (.326) | .436 (.323) | .404 (.356) |
| Average number of physician consultations for children under 1 year old | .283 (.253) | .244 (.237) | .307 (.282) | .227 (.261) |
| **Type of team** | | | | |
| With dental care | .728 | .671 | .703 | .398 |
| With NASF | .448 | .376 | .49 | .194 |
| **Characteristic of the team** | | | | |
| Number of hours of physician per week | 39.1 (7.3) | 38.0 (6.4) | 39.9 (8.3) | 26.4 (18.0) |
| Number of hours of nurse per week | 40.6 (5.3) | 40.2 (3.8) | 40.9 (7.9) | 40.9 (6.9) |
| Number of hours of dentist per week | 30.2 (16.4) | 27.2 (17.8) | 28.8 (17.5) | 16.4 (19.0) |
| Number of hours of nurse assistant per week | 56.5 (28.8) | 51.5 (24.5) | 55.5 (28.0) | 38.3 (31.0) |
| Number of hours of dentist assistant per week | 34.0 (21.8) | 29.3 (20.9) | 31.8 (22.0) | 17.4 (21.1) |
| Number of hours of community health agent per week | 255.9 (78.3) | 266.6 (85.0) | 252.5 (82.1) | 313.1 (179.8) |
| **Socioeconomic status** | | | | |
| Group 1 | .131 | .105 | .117 | .060 |
| Group 2 | .133 | .16 | .143 | .120 |
| Group 3 | .145 | .191 | .160 | .188 |
| Group 4 | .184 | .179 | .184 | .160 |
| Group 5 | .158 | .178 | .173 | .190 |
| Group 6 | .247 | .181 | .219 | .279 |
| **Geographic status** | | | | |
| Rural area | .197 | .303 | .233 | .295 |
| North | .053 | .098 | .070 | .129 |
| Northeast | .321 | .464 | .366 | .438 |
| Midwest | .065 | .076 | .075 | .058 |
| Southeast | .388 | .254 | .337 | .267 |
| South | .170 | .106 | .149 | .106 |
| Small municipality | .149 | .115 | .132 | .066 |
| Median municipality | .475 | .548 | .504 | .487 |
| Large municipality | .375 | .335 | .363 | .445 |

Note: Values are mean (Standard Deviation).

of the teams were below the recommended value of 0.37 consultations per month in the cycle 1, which means 9 consultations for routine growth and development monitoring during the children's first 2 years of live, not including acute care or immunization programmes [22]. In the second cycle, half of the PMAQ teams have achieved the recommended value. The teams at 75th and 90th quantiles performed better than the recommended by the MoH. However, the long upper whisker means that the average number of consultations were quite varied among the quartile group. Furthermore, the cumulative distribution shows that the median remained below the recommended value. The average number of physician consultations for children under 1 year per month had the worst result when compared to the MoH recommendation of

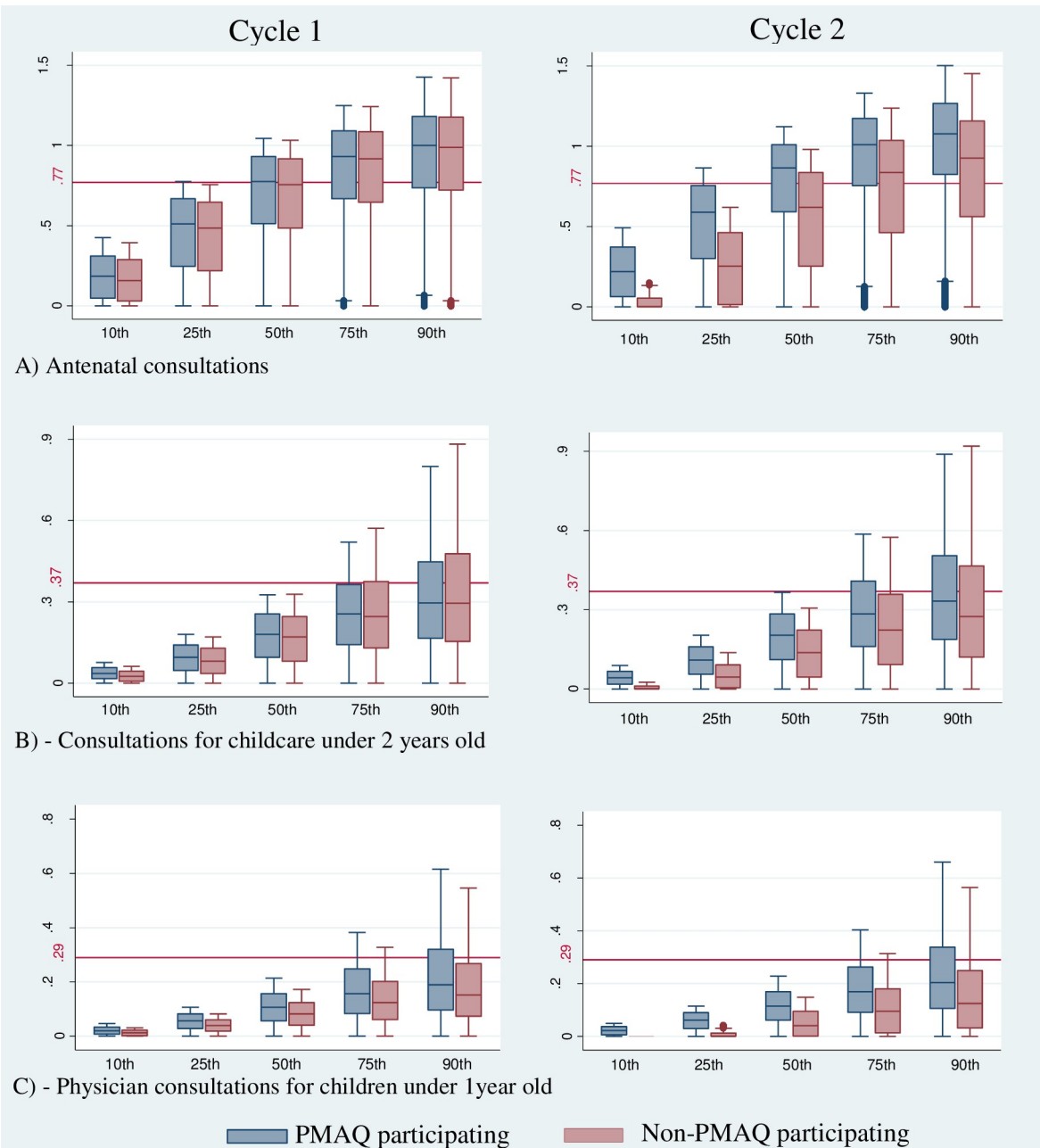

**Fig 1. Cumulative distribution of the average consultations per month for maternal and childcare performed by PMAQ participating and non-participating teams in cycles 1 and 2, Brazil, 2011–2014.** Number of consultations per month recommended by the Ministry of Health (red line). Central line shows median achievement and box shows interquartile range.

0.29 consultations per month, representing 3.5 routine follow-up consultations with a physician per year [22]. The box plot is comparatively short in the lower tail (10th and 25th quantiles). This suggests that, overall, teams had a very low average number of consultations per month. Only 39.5% of PMAQ teams have achieved the recommended number of consultations in cycle 2. Despite the increase in the average number of consultations, delivery of maternal and childcare remains insufficient for all indicators in the lower tail of distributions (Fig 1).

**Table 2. Results from the OLS and QR models for antenatal consultations in the 2nd cycle of PMAQ, Brazil.**

| Variable | PMAQ Cycle 2 | | | | | |
|---|---|---|---|---|---|---|
| | **OLS** | **10th** | **25th** | **50th** | **75th** | **90th** |
| PMAQ participating | .0990*** (.0060) | .1378*** (.0114) | .1361*** (.0090) | .0996*** (.0070) | .0649*** (.0069) | .0229*** (.0078) |
| With dental care | .0317*** (.0113) | .0389* (.0204) | .0368** (.0161) | .0352*** (.0125) | .0144 (.0123) | .0134 (.0140) |
| With NASF | .0461*** (.0043) | .1023*** (.0086) | .0708*** (.0068) | .0388*** (.0053) | .0205*** (.0052) | .0059 (.0059) |
| Characteristic of the team | | | | | | |
| hPhysician | .0028*** (.0003) | .0013*** (.0004) | .0039*** (.0004) | .0036*** (.0003) | .0020*** (.0003) | .0015*** (.0003) |
| hNurse | .00001 (.0004) | -.0014** (.0006) | -.0007* (.0004) | .0006* (.0003) | .0007** (.0003) | .0017*** (.0004) |
| hDentist | -.0018*** (.000) | -.0015** (.0006) | -.0024*** (.0005) | -.0021*** (.0004) | -.0011*** (.0004) | -.0010** (.0004) |
| hNurse assistant | .0017*** (.0001) | .0020*** (.0002) | .0023*** (.0001) | .0016*** (.0001) | .0012*** (.0001) | .0007*** (.0001) |
| hDentist assistant | .0013*** (.0002) | .0013*** (.0007) | .0017*** (.0003) | .0015** (.0002) | .0008*** (.0002) | .0005** (.0003) |
| hCommunity Health Agents | -.0004*** (.000) | -.0005*** (.00004) | -.0005*** (.00003) | -.0005*** (.00002) | -.0003*** (.00002) | -.0003*** (.00002) |
| Socioeconomic status | | | | | | |
| Group 1 | .1537*** (.0217) | .1052*** (.0369) | .1630*** (.0291) | .1870*** (.0227) | .1746*** (.0224) | .1460*** (.0254) |
| Group 2 | .0961*** (.0147) | .0600** (.0259) | .1008*** (.0204) | .1381*** (.0159) | .1151*** (.0156) | .0674*** (.0178) |
| Group 3 | .1127*** (.0145) | .0562** (.0256) | .1069*** (.0202) | .1506*** (.0157) | .1303*** (.0155) | .0780*** (.0176) |
| Group 4 | .0414*** (.0139) | -.0001 (.0239) | .0265 (.0189) | .0804*** (.0147) | .0765*** (.0145) | .0531*** (.0164) |
| Group 5 | -.0013 (.0075) | -.0375*** (.0137) | -.0032 (.0108) | .0151* (.0084) | .0045 (.0083) | -.0042 (.0094) |
| Geographic status | | | | | | |
| Rural area | -.0403*** (.0049) | -.0189*** (.0099) | -.0326*** (.0078) | -.0476*** (.0061) | -.0478*** (.0060) | -.0461*** (.0068) |
| North | -.0756*** (.0083) | -.2290*** (.0150) | -.1115*** (.0125) | -.0304*** (.0097) | -.0061 (.0096) | .0292*** (.0109) |
| Midwest | -.0943*** (.0087) | -.2319*** (.0173) | -.1518*** (.0136) | -.0727** (.0106) | -.0168 (.0105) | .0291** (.0119) |
| Southeast | -.3407*** (.0056) | -.6015*** (.0108) | -.4459*** (.0085) | -.2942*** (.0066) | -.2117*** (.0065) | -.1632*** (.0074) |
| South | -.3527*** (.0078) | -.6169*** (.0137) | -.5216*** (.0108) | -.3267*** (.0084) | -.1848*** (.0083) | -.1086*** (.0094) |
| Small municipality | -.1889*** (.020) | -.1940*** (.0345) | -.2100*** (.0272) | -.1873*** (.0212) | -.1766*** (.0209) | -.1580*** (.0237) |
| Median municipality | -.1109*** (.0123) | -.0862*** (.0210) | -.1031*** (.0166) | -.1197*** (.0129) | -.1005*** (.0127) | -.0673*** (.0145) |
| Constant | 1.0292*** (.0198) | .7336*** (.0298) | .7985*** (.0235) | .9913*** (.0183) | 1.2432*** (.0180) | 1.4452*** (.0205) |
| Number of observations (teams) | 32,948 | 32,948 | 32,948 | 32,948 | 32,948 | 32,948 |

Notes: Values are coefficients (Standard Error). NASF: Family Health Support Centre. hPhysician: working-hour by physicians. hNurse: working-hour by nurses. hDentist: working-hour by dentists. hNurse assistant: working-hour by nurse assistants. hDentist assistant: working-hour by dentist assistants. hCommunity Health Agents: working-hour by community health agents.

The average number of antenatal consultations per month was positively associated with PMAQ participating teams (p<0.01 for both models–OLS and QR) (Table 2). From QR we can see the asymmetric effect of PMAQ on the distribution of antenatal consultations, since there is larger effect in the lower tail of the conditional distribution of the response variable than in the upper tail. In terms of comparison, the OLS coefficient was quite similar with the median (50th quantile) from QR. It is worth noting that coefficients are larger in second cycle compared with the first cycle either in the OLS and QR model. Results from the first cycle can be found in the S2 Table.

Regarding the average number of consultations for childcare under 2 years, OLS estimates of PMAQ participation were not statistically different from zero in both cycles (Table 3 and S3 Table). On the other hand, QR showed mixed results. We found a positive association of PMAQ participating in the lower tail (p<0.01) and the median (p<0.05) of the conditional distribution of the response variable. However, the upper tail (75th and 90th quantiles) was negatively associated with the conditional distribution of the response variable, indicating that PMAQ decreased the number of consultations for childcare under 2 years in both cycles. As observed in antenatal consultations, coefficients were also larger in second cycle than the first one (Table 3 and S3 Table). Results from the first cycle can be found in the S3 Table.

**Table 3. Results from OLS and QR models for average number of consultations for childcare under 2 years old in the 2ⁿᵈ cycle of PMAQ, Brazil.**

| Variable | | PMAQ Cycle 2 | | | | |
|---|---|---|---|---|---|---|
| | OLS | 10th | 25th | 50th | 75th | 90th |
| PMAQ participating | -.0058 (.0053) | .0236*** (.0037) | .0297*** (.0044) | .0127** (.0056) | -.0336*** (.0089) | -.0624*** (.0139) |
| Type of team | | | | | | |
| With dental care | .0384*** (.0087) | .0180*** (.0067) | .0280*** (.0079) | .0450*** (.0101) | .0524*** (.0160) | .0417* (.0250) |
| With NASF | .0043 (.0039) | .0176*** (.0028) | .0182*** (.0034) | .0142*** (.0043) | -.0038 (.0068) | -.0215** (.0106) |
| Characteristic of the team | | | | | | |
| hPhisician | .0015*** (.0002) | .0003** (.0001) | .0008*** (.0002) | .0017*** (.0002) | .0024*** (.0004) | .0021*** (.0005) |
| hNurse | .0001 (.0003) | -.0002*** (.0002) | -.0004* (.0002) | .00004 (.0003) | .0006 (.0004) | .0005 (.0007) |
| hDentist | -.0005 (.0002) | -.0007*** (.0002) | -.0007*** (.0002) | -.0007** (.0003) | -.00003 (.0005) | .0017** (.0008) |
| hNurse assistant | .0005*** (.0001) | .0002*** (.0002) | .0006*** (.0001) | .0007*** (.0001) | .0006*** (.0001) | .0004** (.0002) |
| hDentist assistant | -.0005*** (.0002) | .0005*** (.0001) | .0002 (.0001) | -.0003* (.0002) | -.0009*** (.0003) | -.0018*** (.0005) |
| hCommunity Health Agents | -.0005*** (.00001) | -.0001*** (.00001) | -.0003*** (.00001) | -.0005*** (.00002) | -.0007*** (.00003) | -.0009*** (.00005) |
| Socioeconomic status | | | | | | |
| Group 1 | .1019*** (.0163) | .0232* (.0120) | .0689*** (.0143) | .0947*** (.0181) | .1794*** (.0287) | .1776*** (.0449) |
| Group 2 | .1150*** (.0107) | .0125 (.0084) | .0536*** (.0100) | .0865*** (.0127) | .1943*** (.0202) | .2529*** (.0316) |
| Group 3 | .1020*** (.0104) | .0140* (.0084) | .0451*** (.0100) | .0764*** (.0126) | .1728*** (.0200) | .2203*** (.0313) |
| Group 4 | .0544*** (.0095) | .0100*** (.0078) | .0278*** (.0093) | .0312*** (.0117) | .1002*** (.0186) | .1159*** (.0292) |
| Group 5 | .0299*** (.0054) | .0088* (.0045) | .0136** (.0054) | .0110 (.0068) | .0455*** (.0108) | .0708*** (.0170) |
| Geographic status | | | | | | |
| Rural area | .0465*** (.0047) | .0089*** (.0032) | .0196*** (.0038) | .0464*** (.0049) | .0793*** (.0077) | .0715*** (.0121) |
| North | .0289*** (.0082) | -.0637*** (.0052) | -.0550*** (.0061) | .0219*** (.0078) | .11720*** (.0123) | .1628*** (.0193) |
| Midwest | -.0794*** (.0081) | -.0868*** (.0056) | -.1045*** (.0067) | -.0840*** (.0084) | -.0535*** (.0134) | -.0078*** (.0210) |
| Southeast | -.0932*** (.0046) | -.0630*** (.0035) | -.0732*** (.0042) | -.0883*** (.0053) | -.1089*** (.0085) | -.1245*** (.0133) |
| South | -.1428*** (.0059) | -.0932*** (.0035) | -.1227*** (.0053) | -.1410*** (.0067) | -.1601*** (.0107) | -.1735*** (.0162) |
| Small municipality | -.0719*** (.0153) | -.0604*** (.0112) | -.1168*** (.0134) | -.1050*** (.0169) | -.0880*** (.0268) | .0280 (.0419) |
| Median municipality | -.0448*** (.0083) | -.0296*** (.0068) | -.0595*** (.0082) | -.0526*** (.0103) | -.0602*** (.0164) | -.0234 (.0256) |
| Constant | .4947*** (.0137) | .1297*** (.0098) | .2406*** (.0116) | .3987*** (.0147) | .6200*** (.0233) | .9309*** (.0365) |
| Number of observations (teams) | 32,369 | 32,369 | 32,369 | 32,369 | 32,369 | 32,369 |

Notes: Values are coefficients (Standard Error). NASF: Family Health Support Centre. hPhysician: working-hour by physicians. hNurse: working-hour by nurses. hDentist: working-hour by dentists. hNurse assistant: working-hour by nurse assistants. hDentist assistant: working-hour by dentist assistants. hCommunity Health Agents: working-hour by community health agents.

PMAQ participating teams were positively associated with the average number of physician consultations for children under 1 year in the second cycle, but again much larger in the lower tail (10th and 25th quantiles) and median (50th quantile) and declining to a negligible or not statistically different from zero association in the upper tail (90th quantile) (Table 4). Different from the previous outcomes, there is no clear evidence that second cycle gives larger coefficients compared with first cycle (S4 Table). Results from the first cycle can be found in the S4 Table. S5–S10 Tables present our results considering the imputation for missing values. The results were similar to those estimated in the main analysis.

## Discussion

To the best of our knowledge, our study is the first to estimate the association of PMAQ on the entire conditional distribution of the provision of maternal and child at primary health care in Brazil, controlling for several covariates in terms of socioeconomic, geographic and team characteristics. Our findings indicate that PMAQ contributed to increase the provision of maternal

**Table 4. Results from OLS and QR models for the number of physician consultations for children under 1 year old in the 2nd cycle of PMAQ, Brazil.**

| Variable | PMAQ Cycle 2 | | | | | |
|---|---|---|---|---|---|---|
| | OLS | 10th | 25th | 50th | 75th | 90th |
| **PMAQ participating** | **.0094** (.0039) | **.0126*** (.0022)** | **.0125*** (.0028)** | **.0158*** (.0037)** | **.0037 (.0062)** | **-.0228* (.0121)** |
| Type of team | | | | | | |
| With dental care | .0006 (.0073) | .0110*** (.0040) | .0065 (.0050) | .0029 (.0066) | -.0001 (.0112) | -.0180 (.0217) |
| With NASF | -.0004 (.0030) | .002* (.001) | .0029 (.0021) | -.0008 (.0028) | -.0080* (.0047) | -.0109 (.0092) |
| Characteristic of the team | | | | | | |
| hPhisician | .0018*** (.0002) | .0005*** (.0001) | .0014*** (.0001) | .0025*** (.0001) | .0028*** (.0002) | .0014 (.0005) |
| hNurse | .0008*** (.0002) | -.0004*** (.0001) | .0002* (.00001) | .0005*** (.0002) | .0012*** (.0003) | .0023*** (.0006) |
| hDentist | -.0001 (.0002) | -.0003*** (.0001) | -.0004*** (.0002) | -.0003 (.0002) | .0004 (.0003) | .0016** (.0007) |
| hNurse assistant | .0010*** (.0001) | .0004*** (.00003) | .0007*** (.00003) | .0008*** (.0001) | .0011*** (.0001) | .0014*** (.0002) |
| hDentist assistant | .0001 (.0002) | .0003*** (.0001) | .0002*** (.0001) | .00004 (.0001) | -.0003* (.0002) | -.0008* (.0004) |
| hCommunity Health Agents | -.0004*** (.00001) | -.00004 (.00001)*** | -.0001*** (.00001) | -.0003*** (.00001) | -.0004*** (.00002) | -.0005*** (.00004) |
| Socioeconomic status | | | | | | |
| Group 1 | -.0416*** (.0151) | .0267*** (.0073) | .0239*** (.0090) | -.0005 (.0121) | -.0703*** (.0203) | -.1475*** (.0394) |
| Group 2 | -.0537*** (.0098) | .0127** (.0051) | .0006 (.0063) | -.0390*** (.0084) | -.0780*** (.0142) | -.1342*** (.0275) |
| Group 3 | -.0717*** (.0096) | .00434 (.0050) | -.0115* (.0062) | -.0581*** (.0083) | -.0992*** (.0140) | -.1601*** (.0272) |
| Group 4 | -.0940*** (.0094) | -.0097** (.0047) | -.0295*** (.0058) | -.0795*** (.0078) | -.1183*** (.0131) | -.1827*** (.0254) |
| Group 5 | -.0739*** (.0053) | -.0078*** (.0027) | -.0295*** (.0033) | -.0557*** (.0045) | -.0876*** (.0075) | -.1790*** (.0146) |
| Geographic status | | | | | | |
| Rural area | .0202*** (.0031) | .0153*** (.0020) | .0219*** (.0024) | .0254*** (.0032) | .0291*** (.0055) | .0202* (.0106) |
| North | .0714*** (.0049) | -.0014 (.0031) | .0214*** (.0038) | .0568*** (.0051) | .1015*** (.0086) | .1491*** (.0168) |
| Midwest | .1427*** (.0059) | .0319*** (.0034) | .0702*** (.0042) | .1299*** (.0056) | .2000*** (.0094) | .2561*** (.0183) |
| Southeast | .0957*** (.0037) | -.0105*** (.0021) | .01200*** (.0026) | .0602*** (.0035) | .1370*** (.0059) | .2611*** (.0115) |
| South | .1126*** (.0052) | -.0043 (.0027) | .0241*** (.0033) | .0891*** (.0045) | .1907*** (.0075) | .3121*** (.0146) |
| Small municipality | .0456*** (.0145) | .0049 (.0068) | .0089 (.0084) | .0323*** (.0113) | .0879*** (.0190) | .0909*** (.0368) |
| Median municipality | .0034 (.0084) | .0030 (.0041) | -.0002 (.0051) | .0138** (.0068) | .0123 (.0115) | -.0197 (.0223) |
| Constant | .2233*** (.0119) | .0170*** (.0058) | .0321*** (.0072) | .1081*** (.0097) | .2382*** (.0163) | .5189*** (.0316) |
| Number of observations (teams) | 33,180 | 33,180 | 33,180 | 33,180 | 33,180 | 33,180 |

Notes: Values are coefficients (Standard Error). NASF: Family Health Support Centre. hPhysician: working-hour by physicians. hNurse: working-hour by nurses. hDentist: working-hour by dentists. hNurse assistant: working-hour by nurse assistants. hDentist assistant: working-hour by dentist assistants. hCommunity Health Agents: working-hour by community health agents.

and childcare, particularly in teams at the lower tail (10th and 25th quantiles) of the conditional distribution of the response variables. This means that teams with lower number of antenatal or child consultations benefited the most by participating in PMAQ, which suggest that PMAQ might motivate worse performing health providers to catch up. Our results seem to be consistent with previous studies dealing with P4P strategies at primary health care. Das et al. [42] reported positive but modest effects on process indicators related to maternal care in low- and middle-countries. Moreover, systematic reviews have showed that P4P strategies enhance improvements in areas where the baseline performance was low [43, 44]. However, Engineer et al. [45] found no statistically significant results on provision of maternal and childcare due to P4P strategies in Afghanistan.

PMAQ was implemented to improve access and quality for several health conditions besides the provision of maternal and childcare, such as cancer screening, management of diabetes and hypertension, disease prevention and health promotion. On this basis, it is expected that when teams achieve more consultations than recommended by the MoH they should

decrease their provision of care. This seems to be the case of negative or not statistically different from zero coefficients at upper tail for childcare (75th and 90th quantiles). However, it is worth noting that we used aggregate data (average number of consultations per month), which do not allow us to know which children were being cared for. It is possible that a few children are attending consultations excessively, leaving others out. Unfortunately, individual data are not available at national level to shed light on this question.

Many countries have recommended normative standards to improve good practice in maternal and childcare, particularly in terms of number of consultations [21–23, 46]. These normative standards are important since they would represent the users' potential demand or health need. The Brazilian MoH recommends 7 antenatal contacts during the pregnancy [21] and the World Health Organization (WHO) recommends 8 contacts [23]. In the first cycle, PMAQ participating teams at 10th and 25th quantiles did not achieve these parameters recommended by MoH or WHO, while the teams at 75th and 90th quantiles performed 1.6 and 1.8 times higher than the recommendations, respectively. In the second cycle, PMAQ participating teams have increased the number of antenatal consultations in the entire distribution, and just teams at 10th quantile did not achieve the recommendations. In both cycles, PMAQ participating teams at 10th quantile achieved around half of the recommended parameters. The results were worse for the other process outcomes investigated. For example, only teams at the upper tail (75th and 90th quantiles) have achieved the MoH's recommendation of 9 consultations for children under 2 years [22] and the parameter of 3.5 physician consultations for child under 1 year [46]; at 10th quantile, PMAQ participating teams achieved 20% of the recommendation for children under 2 years, and 10% of the physician consultations for child under 1 year. Other studies also reported similar results, showing that among 67% and 73% of pregnant women received 6 to 7 antenatal consultations in Brazil [17, 25, 47]. A study carried out in the South and Northeast of Brazil showed that only 25% of the children under 2 years received 9 or more consultations in the first two years of life [48].

França et al. [49] found a reduction on socioeconomic inequalities related to maternal care, which can be associated with the initial expansion of FHS in more vulnerable areas in Brazil. Our results showed that PMAQ participating teams from the poorest municipalities (groups 1, 2, 3) have increased the average number of antenatal consultations per month (p<0.05) compared to the richest municipalities (group 6). We also found evidence on the number of consultations for children under 2 years, where poorest municipalities (groups 1 and 2) had positive coefficient (p<0.01) on the response variable in the first (all quantiles investigated) and second (25th, 50th, 75th and 90th quantiles) cycles. David et al. [50] and Santos et al. [51] also reported better performance at primary healthcare in the poorest municipalities. Worse performance at primary healthcare in the richest municipalities may be associated with higher private health insurance coverage, supply of inpatient and outpatient facilities (secondary, tertiary and quaternary care), and lower primary healthcare coverage [52–54].

The community health agent workforce was associated with a decreased average number of maternal and childcare consultations per month (p<0.01) in both cycles, while the nurse assistant workforce was positively associated with the outcomes, for which most of the coefficients were statistically significant at p<0.01. Although the community health agents represent the link between the health facility and the catchment population, some challenges remain such as the misperception about their role within FHT, poor work-conditions, obstacles in the relationship with the community and teams, weak professional training, and bureaucracy [55]. There are few studies about the services delivered by community health agents at national level in Brazil. A local study, which analysed data from 121 community health agents in ten municipalities, showed that 41.3% and 19.0% of the sample provided care to pregnant women and children, respectively [56]. The most frequent activities were home visit (90,1%) and situation analysis of the catchment families (95.0%), both in terms of health problems and vulnerability

[56]. This may suggest the maternal and childcare are not a high-priority group for community health agents, since they have many other duties. Moreover, nurses seem to have high perceived-impact of PMAQ while the community health agents a low perceived-impact [57] which could explain more knowledge about PMAQ by the nurses, and consequently more engagement with the programme.

Findings from a qualitative study carried out in the south of Brazil showed that PMAQ contributed to better organization and registration of information, to plan health actions, and to mobilize health team and management to meet targets established in the program [58]. This could explain the cases where PMAQ teams have improved performance without increasing hours of their doctors, nurses etc.

NASF seems to play a role in the provision of maternal and childcare at least in the lower tail of the conditional distribution of outcome variables (10th and 25th quantiles). This positive association is larger for the average number of antenatal consultations per month and in the second cycle, when NASF achieved 50% coverage in PMAQ participating teams. There is evidence that NASF may strengthen the assistance coordination of primary healthcare, mainly related to information coordination and clinical management [59]. However, some authors have argued that the work done by NASF professionals must be better integrated within the FHS, as well as it is necessary to deepen the understanding of results that can actually be attributed to the NASF [60]. We did not identify any study that analysed the link between PMAQ and NASF.

We addressed several problems that we have identified in the previous studies. First, we used a quantile regression model to estimate the association of covariates on the conditional distribution of the outcome variables, since the previous studies were essentially descriptive analyses. Second, we considered a broad set of covariates in order to control several factors such as socioeconomic, geographical and team characteristics. Third, we analysed all teams at national level, apart from 15 to 20% of FHTs dropped due to missing data. However, similar results were obtained when we employed a method for multiple imputation.

Several limitations of our study should be acknowledged. First, we used administrative data from the MoH, which presented some inconsistences related to identification of teams, particularly in the first cycle when the INE was implemented. Second, SIAB is a database based on aggregate information, which do not allow us to get user-level data. Third, we sought to control for other covariates such as education and income per capita at team level, as well as other programmes launched during the PMAQ implementation such as *Rede Cegonha* (Stork Network, in English) which is a national strategy to improve pregnancy and newborn healthcare, but these data were not available. Fourth, another potential source of bias is that not all maternal and child care were captured in our study e.g. acute care and immunization programmes. Based on that, our results may underestimate the role of PHT in delivering maternal and child care. We investigated only antenatal and consultations for routine growth and development monitoring during the children's first 2 year. Moreover, increasing provision of care may not necessarily mean increasing quality of care. Fifth, we used a cross sectional approach to perform our estimates, and further investigation should focus on causality. Sixth, PMAQ may be implemented in many different ways across the country, so it is not possible, based on national databases, to know how much incentive was received at team level, if any. Moreover, it is important to inform the MoH whether the PMAQ as a whole is associated with better results than non-PMAQ participating teams in terms of maternal and childcare.

## Conclusions

PMAQ is associated with the increased provision of care to pregnant women and children under 2 years old at primary healthcare level. Teams with lower average number of antenatal

or child consultations per month benefited the most by participating in PMAQ, since they presented larger positive coefficients in the quantile regression models. However, this was not enough to overcome the very low number of consultations at 10th and 25th quantiles taking into consideration the MoH guidelines.

As P4P are conceived to be an open approach adapted to specific needs, our findings can help the MoH to foster worse performers (teams at 10th and 25th quantiles) to catch up. In this context, Brazil's experience demonstrates that P4P can be a driver for improving equity on the provision of maternal and childcare.

Our findings require further investigation on quality of care, since we focused on quantity of consultations for maternal and childcare. Although this is the first attempt to evaluate the PMAQ against its impact on maternal and childcare provision, the PMAQ scope is larger than that which we have investigated in this study. There are opportunities to explore impact also in other clinical domains such as diabetes and hypertension management, immunization coverage, health education programmes and working conditions.

## Supporting information

**S1 Table. Description of variables used in the analysis.**
(DOCX)

**S2 Table. Results from the OLS and QR models for antenatal consultations in the 1st Cycle of PMAQ, Brazil.**
(DOCX)

**S3 Table. Results from OLS and QR models for average number of consultations for childcare under 2 years old in the 1st Cycle of PMAQ, Brazil.**
(DOCX)

**S4 Table. Results from OLS and QR models for the number of physician consultations for children under 1 year old in the 1st Cycle of PMAQ, Brazil.**
(DOCX)

**S5 Table. Results from the OLS and QR models for antenatal consultations in the 1st Cycle of PMAQ (missing values imputed), Brazil.**
(DOCX)

**S6 Table. Results from OLS and QR models for average number of consultations for childcare under 2 years old in the 1st Cycle of PMAQ (missing values imputed), Brazil.**
(DOCX)

**S7 Table. Results from OLS and QR models for the number of physician consultations for children under 1 year old in the 1st Cycle of PMAQ (missing values imputed), Brazil.**
(DOCX)

**S8 Table. Results from the OLS and QR models for antenatal consultations in the 2nd Cycle of PMAQ (missing values imputed), Brazil.**
(DOCX)

**S9 Table. Results from OLS and QR models for average number of consultations for childcare under 2 years old in the 2nd Cycle of PMAQ (missing values imputed), Brazil.**
(DOCX)

**S10 Table. Results from OLS and QR models for the number of physician consultations for children under 1 year old in the 2nd Cycle of PMAQ (missing values imputed), Brazil.** (DOCX)

## Author Contributions

**Conceptualization:** Olívia Lucena de Medeiros, Jorge Otávio Maia Barreto, Everton Nunes da Silva.

**Data curation:** Olívia Lucena de Medeiros, Letícia Xander Russo.

**Formal analysis:** Olívia Lucena de Medeiros, Letícia Xander Russo, Everton Nunes da Silva.

**Investigation:** Olívia Lucena de Medeiros.

**Methodology:** Olívia Lucena de Medeiros, Everton Nunes da Silva.

**Supervision:** Jorge Otávio Maia Barreto, Everton Nunes da Silva.

**Validation:** Matthew Harris, Letícia Xander Russo, Everton Nunes da Silva.

**Writing – original draft:** Olívia Lucena de Medeiros, Everton Nunes da Silva.

**Writing – review & editing:** Jorge Otávio Maia Barreto, Matthew Harris, Letícia Xander Russo, Everton Nunes da Silva.

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
