## [Decision Letter · Decision Letter 0]

29 Jun 2020

PONE-D-20-14125

Delivering maternal and childcare at primary healthcare level: the role of PMAQ as a pay for performance strategy in Brazil

PLOS ONE

Dear Dr. da Silva,

Thank you for submitting your manuscript to PLOS ONE. After careful consideration, we feel that it has merit but does not fully meet PLOS ONE’s publication criteria as it currently stands. Therefore, we invite you to submit a revised version of the manuscript that addresses the points raised during the review process.

We suggest minor revisions based on reviewer's comments. We also suggest to reply to reviewer#3 explaining the choice and validity of the statistical analysis. The others reviewers did not comment on that and we  believe your feedback should be enough.

We look forward to receiving your revised manuscript.

Kind regards,

Elena Ambrosino

Academic Editor

PLOS ONE

Journal Requirements:

3. Please include your tables as part of your main manuscript and remove the individual files. Please note that supplementary tables (should remain/ be uploaded) as separate "supporting information" files

Additional Editor Comments (if provided):

Reviewers' comments:

Reviewer's Responses to Questions

**Comments to the Author**

1. Is the manuscript technically sound, and do the data support the conclusions?

Reviewer #1: Yes

Reviewer #2: Yes

Reviewer #3: Partly

Reviewer #4: Partly

2. Has the statistical analysis been performed appropriately and rigorously? 

Reviewer #1: Yes

Reviewer #2: Yes

Reviewer #3: I Don't Know

Reviewer #4: I Don't Know

3. Have the authors made all data underlying the findings in their manuscript fully available?

Reviewer #1: Yes

Reviewer #2: Yes

Reviewer #3: Yes

Reviewer #4: Yes

4. Is the manuscript presented in an intelligible fashion and written in standard English?

Reviewer #1: Yes

Reviewer #2: Yes

Reviewer #3: No

Reviewer #4: Yes

5. Review Comments to the Author

Reviewer #1: Over the last decade, there has been an increase in the use of pay for performance (P4P) strategies as an approach for improving availability and utilization of healthcare services. In this article, authors Medeiros et al. explore the impact of (P4P) at a national level throughout Brazil on the provision of maternal and child health care. Specifically, the authors use a cross sectional quantile regression analysis to assess the association between P4P, or the strategy of incentivizing providers to delivery key services, and the provision of key maternal and child health services across national Family Health Teams. Authors explored the association over time, analyzing 33,368 Family Health Teams between 2011-2012 (Cycle 1) and 39,211 teams between 2013-2014 (Cycle 2). Key findings from the study include: a positive association between P4P participation and antenatal visits per month, with an increased effect in lower quantiles; a positive association between P4P participation and the number of visits for children under two per month in the lower quantiles; and a positive association between P4P and visits for children under one in the lower tail. Results also indicated that there were considerable portions of the teams that did not achieve national recommendations for maternal and child health visits. The current manuscript requires some additional edits and considerations for added context; please see below for specific suggestions.

(1) This comes up a bit in the discussion, through the comparison of findings to existing literature, but it may strengthen the argument for the need for this study to provide – upfront in the introduction – a discussion of any ethical considerations associated with a P4P system. For example, do increased visits under P4P lead to increased quality of care or improved outcomes? The data used in the study does not allow the authors to identify an association between P4P and clinical outcomes, but consider including a more comprehensive review of the P4P literature in the introduction, rather than the majority of this literature residing in the discussion section.

(2) It may be useful to the reader for the authors to provide a more detailed discussion of how covariates for the model were selected. The manuscript provides an appropriate description of the indicators included in the model, but providing additional detail and context on selection would be helpful. For example, were the covariates selected based on theory? Previous research? Existing literature?

Additional, more minor areas where it would be helpful to address edits include:

(1) For clarity purposes, it may be helpful to add an additional sentence or two describing the structure of the Family Health Teams. For readers unfamiliar with the Brazil health system, the relationship between Family Health Strategy and Primary Health Teams within Family Health Teams is likely unknown.

(2) Acronyms are not consistent throughout the manuscript. There are instances where acronyms are previously defined, and then the full term is used later on throughout the paper. This was most commonly seen for pay for performance (P4P), Ministry of Health (MOH), and Family Health Strategy (FHS).

(3) At the end of the introduction, the authors describe the key outcomes being measured in the analysis, indicating that they were selected for clinical purposes, as they are essential for good practice in MCH. Consider providing a few additional sentences more clearly linking the selected outcomes with clinical outcomes, based on what is known in the literature, to clarify to the reader why these were selected.

(4) Within the Participants section of the manuscript, the authors describe the fact that there was significant data missing from the dataset used for both cycles. However, this does not seem to be mentioned in the discussion as a limitation. Consider discussing the potential impact of this missing data as a limitation.

(5) Finally, this manuscript would benefit from an overall copy edit to address any minor spelling or grammatical errors.

Reviewer #2: This is an interesting study, with the limitations that the Authors well acknowledged in Discussion. I have no idea of neonatal death numbers in Brazil, and whether these number would change as a result of P4P, at least in the most disadvantaged areas.

Reviewer #3: This paper addresses an important topic which the impact of the pay for performance policy on access to quality MCH services.

My main concern is about the statistical methods used to analyze the data. I am not familiar with those methods and I suggest that PLoS One hires a statistician to review that part of the paper.

Reviewer #4: This manuscript represents a comparative analysis of maternal and child service utilization of a pay for performance program for primary care providers in Brazil from 2001 - 2004. Overall, this is a well written manuscript and the information is timely given the expansion of pay for performance programs. However, there are some gaps in the methods that should be mentioned earlier or adjusted in the analysis, specifically overlap in outcome measures for child care visits at both one and two years and the potential omission of immunization visits. The manuscript would also benefit from review by a native English speaker for a few areas with subject-verb agreement issues or incorrect work choice or syntax.

Introduction:

• The Introduction is well-reasoned and provides insight into the context. Of note, the Introduction indicates that the PMAQ third cycle still had not achieved complete coverage -

Methods:

• The description of the analysis is fairly complete but it is unclear why the authors selected the dates ranges for time as remote as 2014 – it is now six years later and it would be helpful to see longer-term trends, assuming this program continued. Please add detail for why these time bands were selected and why the analysis was limited to only two cycles.

• The outcomes of child care visits for children under one year and a separate variable for children under two years reflect overlap and would likely be collinear if used in the same models.

• This is a minor point but I would be careful regarding the use of the word “dummy” for variables. It is reasonable to categorize variables for expediency as long as the categories are defined and they do not need to be referred to as “dummy variables” unless they represent multiple measures within one variable, which is not the case here.

Results:

• Figure 1 would benefit from clear labeling of all content – please remember the adage that figures and tables should be able to stand on their own with needing text to describe the findings.

• In the tables, please limit results to only 1 decimal point in Table 1 and no more than three decimal places in Tables 2-4. The sheer volume of numbers make the data very difficult to interpret.

• I question whether including dental services and providers or the mental health teams is worthwhile in this analysis – how do the authors see this impacting MCH service utilization? More parsimonious models would be preferable.

• The same question asked earlier about overlap between children under 1 and children under 2 is more prominent here and the reader is left wondering what the explanation is for such a difference between these two outcomes with respect to coverage.

Discussion:

• Overall, the Discussion is comprehensive but the Limitations section mentions a major limitation that should be raised in Methods. Immunization visits are often the bulk of well-child visits, particularly in the first year. Having these visits potentially excluded from the analysis may result in substantial bias and should be mentioned in Methods as well.

• The mention that community health agent workforce was associated with fewer visits deserves greater attention as this seems counter-intuitive – was there any care provided at community level that may have resulted in not visiting the facility?

6. PLOS authors have the option to publish the peer review history of their article (what does this mean?). If published, this will include your full peer review and any attached files.

Reviewer #1: No

Reviewer #2: No

Reviewer #3: **Yes: **Blami Dao

Reviewer #4: No

---

## [Author Response · Author response to Decision Letter 0]

4 Aug 2020

We attached the response to reviewers in the submission system.

---

## [Decision Letter · Decision Letter 1]

20 Aug 2020

PONE-D-20-14125R1

Delivering maternal and childcare at primary healthcare level: the role of PMAQ as a pay for performance strategy in Brazil

PLOS ONE

Dear Dr. da Silva,

Thank you for submitting your manuscript to PLOS ONE. After careful consideration, we feel that it has merit but does not fully meet PLOS ONE’s publication criteria as it currently stands. Therefore, we invite you to submit a revised version of the manuscript that addresses the points raised during the review process. .

It's important that the points raised by the reviewers are taken into consideration and incorporated in the revised manuscript as much as possible to improve its quality. Please make sure to do so in the next revision.

We look forward to receiving your revised manuscript.

Kind regards,

Elena Ambrosino

Academic Editor

PLOS ONE

Reviewers' comments:

Reviewer's Responses to Questions

**Comments to the Author**

1. If the authors have adequately addressed your comments raised in a previous round of review and you feel that this manuscript is now acceptable for publication, you may indicate that here to bypass the “Comments to the Author” section, enter your conflict of interest statement in the “Confidential to Editor” section, and submit your "Accept" recommendation.

Reviewer #3: All comments have been addressed

Reviewer #4: (No Response)

2. Is the manuscript technically sound, and do the data support the conclusions?

Reviewer #3: Yes

Reviewer #4: Partly

3. Has the statistical analysis been performed appropriately and rigorously? 

Reviewer #3: Yes

Reviewer #4: No

4. Have the authors made all data underlying the findings in their manuscript fully available?

Reviewer #3: Yes

Reviewer #4: Yes

5. Is the manuscript presented in an intelligible fashion and written in standard English?

Reviewer #3: Yes

Reviewer #4: Yes

6. Review Comments to the Author

Reviewer #3: Can the author do the following changes to the manuscript?

- in the results section, instead of describing the results by Figure 1 shows.., table 1 shows.., he should make a sentence and put the figure or table number into brackets

- at the end, instead of final considerations, he/she should put conclusions

Reviewer #4: While I appreciate the time taken by the authors to write responses to the points raised in review, very few of the suggestions to improve the manuscript were taken on board. This lack of responsiveness in the manuscript occurred even when the authors agreed with the point raised, such as excluding immunization visits, which should be mentioned more prominently as a source of bias and potential care at the community level, which should be raised in the Discussion. Reviewers spend quite a bit of time as volunteers to evaluate and recommend ways to improve manuscripts. In my view, the authors need to integrate more of these recommendations, particularly surrounding the analysis, potential biases, presentation of values in the tables, and quality of the writing, before this manuscript can be accepted.

7. PLOS authors have the option to publish the peer review history of their article (what does this mean?). If published, this will include your full peer review and any attached files.

Reviewer #3: **Yes: **Blami Dao

Reviewer #4: No

---

## [Decision Letter · Decision Letter 2]

30 Sep 2020

Delivering maternal and childcare at primary healthcare level: the role of PMAQ as a pay for performance strategy in Brazil

PONE-D-20-14125R2

Dear Dr. da Silva,

We’re pleased to inform you that your manuscript has been judged scientifically suitable for publication and will be formally accepted for publication once it meets all outstanding technical requirements.

Kind regards,

Elena Ambrosino

Academic Editor

PLOS ONE

Additional Editor Comments (optional):

Reviewers' comments:

Reviewer's Responses to Questions

**Comments to the Author**

1. If the authors have adequately addressed your comments raised in a previous round of review and you feel that this manuscript is now acceptable for publication, you may indicate that here to bypass the “Comments to the Author” section, enter your conflict of interest statement in the “Confidential to Editor” section, and submit your "Accept" recommendation.

Reviewer #4: All comments have been addressed

2. Is the manuscript technically sound, and do the data support the conclusions?

Reviewer #4: Yes

3. Has the statistical analysis been performed appropriately and rigorously? 

Reviewer #4: Yes

4. Have the authors made all data underlying the findings in their manuscript fully available?

Reviewer #4: Yes

5. Is the manuscript presented in an intelligible fashion and written in standard English?

Reviewer #4: Yes

6. Review Comments to the Author

Reviewer #4: (No Response)

7. PLOS authors have the option to publish the peer review history of their article (what does this mean?). If published, this will include your full peer review and any attached files.

Reviewer #4: **Yes: **Catherine Todd

---

## [Editor Report · Acceptance letter]

5 Oct 2020

PONE-D-20-14125R2 

Delivering maternal and childcare at primary healthcare level: the role of PMAQ as a pay for performance strategy in Brazil 

Dear Dr. da Silva:

I'm pleased to inform you that your manuscript has been deemed suitable for publication in PLOS ONE. Congratulations! Your manuscript is now with our production department. 

Kind regards, 

on behalf of

Dr. Elena Ambrosino 

Academic Editor

PLOS ONE